# Guided Endodontics as a Personalized Tool for Complicated Clinical Cases

**DOI:** 10.3390/ijerph19169958

**Published:** 2022-08-12

**Authors:** Wojciech Dąbrowski, Wiesława Puchalska, Adam Ziemlewski, Iwona Ordyniec-Kwaśnica

**Affiliations:** 1Department of Dental Prosthetics, Faculty of Medicine, Medical University of Gdansk, 80-210 Gdansk, Poland; 2Department of Paediatric Dentistry, Faculty of Medicine, Medical University of Gdansk, 80-210 Gdansk, Poland; 3Private Practice Impladent Medical & Dental Clinic, 80-125 Gdansk, Poland

**Keywords:** guided endodontics, computer-aided design, computer-aided manufacturing, digital imaging, pulp canal obliteration, endodontic guide, root canal treatment

## Abstract

The aim of this paper is to present a technique to individualize root canal localization in teeth with calcified root canals using a digitally planned, 3D-printed endodontic guide. Root canal calcification is characterized by the apposition of tertiary dentin along the canal wall. The endodontic treatment of teeth with calcified canals is often challenging. However, digital dentistry meets these challenges. Merging CBCT images with an intraoral scan allows a clinician to prepare an endodontic guide. This article describes the clinical and digital workflow of the guided endodontic access approach in teeth with difficulties in terms of root canal localization due to post-traumatic pulp canal obliteration (PCO) and canal calcification in elderly patients. The path of entry into the root canal system was planned using cone-beam computed tomography (CBCT). The template was printed on a 3D printer using transparent resin. During root canal treatment (RCT), the endodontic tool was inserted through the sleeve until the desired location was reached. The use of an endodontic guide allowed for minimally invasive RCT, avoiding the excessive loss of tooth structures. Navigated endodontics enables clinicians to perform RCT in a more predictable manner and allows clinicians to avoid iatrogenic complications, which improves the treatment prognosis.

## 1. Introduction

Over the last three decades, there have been many technological advancements. Among the new advancements used in clinical dentistry are 3D printers and 3D cone-beam computed tomography (CBCT) technology. Guided surgery was originally introduced in neurosurgery to perform safe and predictable brain surgeries in a minimally invasive manner [1]. Subsequently, the method was applied to other fields of medicine. For many years, templates for implant insertions have been successfully used in accordance with the previously planned, and prosthetically and anatomically optimal positions (prosthetically driven implantology) [2]. Years of experience in guided surgery have shown the great accuracy of teeth-supported templates, which may be used in endodontics [3].

Among the treatment objectives of root canal treatment (RCT) is the need to preserve normal periradicular tissues. RCT is performed by removing pulp, precise shaping, and cleaning the root canal with the aim of eliminating microorganisms in the root canal system [4].

After evaluating medical and dental histories, determining the chief complaint, and diagnosing a need for RCT, the clinician needs to properly access the tooth. Minimally invasive cavity access preparation in endodontics was presented by Clark and Khademi with the aim of preserving the pericervical dentin and bank tooth structure as a future asset. Preserving the pericervical dentin functions as a stress distributor. The pericervical dentin may improve the resistance to the fracture and increase the probability of success with a future prosthetic restoration [5,6].

The endodontic treatment of teeth with pulp canal obliteration (PCO) is often a significant challenge. It can involve much time and energy and be complicated by difficulties in cavity access, massive hard tissue loss, or perforation—even when performed with the help of magnifying devices [7,8,9,10].

PCO is characterized by the apposition of hard tissue (tertiary dentin) along the root canal wall, which results in a reduction in pulp space volume and root canal diameter. Tertiary dentin is deposited in response to injury (e.g., traumatic injuries, caries, orthodontic therapy, etc) as a result of the pulp healing process. It may lead to partial or complete PCO [7,8,11]. PCO is common in young permanent teeth following luxation injuries, though the frequency of PCO varies in the clinical literature from 3.7% to 40% after traumatic dental injuries (TDI) and from 29.4% to 95.2% after root fractures [9,12,13,14]. Root canal calcification may also be caused by pulp aging processes due to the lifelong physiological apposition of secondary and tertiary dentine. The number of elderly patients who need RCT is increasing [15]. PCO may result in crown discolouration and lower or negative responses to pulp sensibility tests, although these criteria do not allow clinicians to define pulp necrosis after PCO. However, tooth discolouration, even without pulp necrosis, may require endodontic therapy to help aesthetic treatment. The diagnosis of pulp necrosis is based on clinical symptoms (e.g., pain, tenderness to percussion, etc.) and radiological changes in the periapical area and requires endodontic treatment [10,16,17]. Krastl et al. [18] presented a digitally guided method to localize obliterated canals using CBCT and digital impressions/intraoral scans for the endodontic guide. It was designed to facilitate the localization of canals and may allow clinicians to decrease hard tissue removal and reduce chair time [15,19].

After qualifying the tooth for the guided root canal treatment, CBCT imaging and an intraoral scan should be performed (Figure 1). In the computer-aided design stage, the DICOM (Digital Imaging and Communication in Medicine) and STL (standard triangle language) files are imported into digital planning software.

The CBCT image allowed us to localize the visible part of the root canal, define the position of the virtual implant (virtual image of the drill), and plan the access path to the root canal system while preserving the pericervical dentine (Figure 2) [15,20,21]. The dimensions of the implant (drill), such as diameter and length, should match the dimensions of the tool that the clinician uses to access the root canal.

The next step was to align the DICOM and STL files to design a teeth-supported template (Figure 3).

A teeth-supported guide provides good stabilization, eliminates trauma to the gingiva, and allows a clinician to use a rubber dam during the procedure. The guide should cover the labial and palatal surfaces of the three adjacent teeth to secure correct intraoral stabilization (Figure 4). The height of the built-in sleeves should be adapted to the working length of the tools to reveal the visible part of the canal [3].

In the computer-aided manufacturing stage, the STL file of the template project was exported from the planning software and processed in slicer software. This software (Chitubox) allowed us to prepare the file for 3D printing by adding supports and slicing the object into 50 µm layers (Figure 5).

The guide should be printed from resin dedicated for surgical guides—a class IIa product that can be sterilized and safely used in the oral cavity (Figure 6) [22]. The entire digital workflow is presented in Figure 7.

The aim of this paper is to present a clinical and digital workflow of guided endodontics based on four cases performed in an endodontic office between December 2021 and April 2022.

## 2. Case Series Presentation

### 2.1. Post-Traumatic Pulp Canal Obliteration (PCO)

#### 2.1.1. Case Report—Patient A: Symptomatic Pulp Necrosis after Trauma

A 21-year-old female patient reported to the endodontic specialist complaining of moderate pain upon percussion of her maxillary central left incisor (tooth 21), experienced for several weeks. The patient was healthy and had no general or chronic diseases. Her dental history revealed that she had experienced dental trauma a few years earlier. The clinical examination revealed a slightly discoloured maxillary left central incisor, tenderness to percussion, a negative response to the pulp sensibility test (cold test), and moderate pain during soft tissue palpation in the apical region of the maxillary central left incisor. Radiographic images revealed the pulp canal calcification and an absence of the canal light that extended to the middle. A CBCT scan was performed (CS 8100 3D, Carestream, 50 × 50 mm) that confirmed canal calcification. The root canal lumen was visible 7.58 mm from the apex (Figure 8). Due to the possible risk of higher tooth substance loss and perforation during endodontic access, we decided to perform the endodontic treatment with guided access.

##### Three-Dimensional Treatment Plan

The upper arch was scanned with an intraoral scanner (PrimeScan, Dentsply Sirona) instead of analogue impressions. The STL file of the upper arch and the CBCT images were combined in Blue Sky Plan (Blue Sky Bio). The accuracy of the merging was cross-checked in three dimensions (Figure 9). An individual virtual implant with a diameter of 1.5 mm was used during planning, in accordance with the diameter of the endodontic access tool (Figure 10). The CBCT images allowed us to plan the position of the virtual implant so that the drill apex was located at the top of the visible part of the root canal system [23] (Figure 11).

After the implant position was accepted by a clinician, an endodontic guide (teeth-supported) was designed for a sleeveless, static navigation of the image obtained from an intraoral scan (Figure 12). The guide covered the labial and palatal surfaces of the adjacent teeth (13-23) to obtain adequate intraoral stabilization. The use of stabilizing pins is unnecessary in such cases. The guide tube was designed so that the top of the sleeve was 21 mm from the radiographically visible root canal lumen. The guide sleeve was 6 mm long and 1.5 mm in diameter. The endodontic guide was printed on a resin 3D printer (Phrozen Sonic Mini 4K) using a transparent resin designed for printing surgical guides (NextDent SG)—a class IIa product that can be used in the oral cavity (Figure 13). Post-processing was conducted in accordance with the resin manufacturer’s instructions to avoid dimensional changes [24]. A master model was printed to check the accuracy of the guide’s fitting (Figure 14).

##### Canal Treatment

The correct fit of the guide was verified before and after the rubber dam was placed. For local anaesthesia, 1.7 mL Ubistesin forte (3 M ESPE, Seefeld, Germany) was administered (articaini hydrochloridum 40 mg + epinephrine hydrochloridum 0.012/1 mL). A small sign was made on the enamel surface through the guide to indicate the access point. The enamel was removed in a minimally invasive manner with a diamond bur until the dentine was exposed. The guide was placed on the teeth and the treatment was performed with Munce Discovery Bur #1 (CJM Engineering, Santa Barbara, CA, USA) with a speed of 10,000 rpm. This bur was a round carbide bur with a head diameter of 0.8 mm (ISO head size 08) and a total length of between 31 and 34 mm. The guide was removed every 2 mm to rinse the cavity, control endodontic access using an optical microscope, and clean the bur. After the bur reached the estimated depth, the C-Pilot #10 and #15 files (VDW, Munich, Germany) were used to check the canal position. Radiographic examination was performed to confirm correct canal access (Figure 15). The canal was reached at a length of 22 mm from the top of the guide sleeve - 1 mm deeper than the virtually planned depth. Once the remaining canal was reached, the length was confirmed with an electronic apex locator (Raypex 6; VDW, Munich, Germany). A conventional root canal treatment followed. The canal was irrigated with 5.25% sodium hypochlorite (NaOCl), including passive ultrasonic activation, and instrumented with ProTaper Gold (Dentsply Sirona Endodontics, Ballaigues, Switzerland) up to the F3 file size (0.30/0.09v). It was dried with paper points and obturated with vertically condensed gutta-percha and epoxy sealer (AH Plus, De Trey, Konstanz, Germany). The access cavity was cleaned and filled with a composite resin (Estelite Asteria, Tokuyama Dental Corporation, Tokyo, Japan).

#### 2.1.2. Case Report—Patient B: Asymptomatic Pulp Canal Obliteration after Trauma in Tooth Requiring Prosthetic Treatment (Aesthetic Dental Crown)

A 39-year-old male patient was referred to an endodontic specialist for the treatment of a maxillary left central incisor (tooth 21) before prosthetic treatment. Endodontic treatment had been initiated by the referring dentist, but as it was impossible for the clinicians to localize the root canal under the optical microscope, the treatment could not be completed, and the patient was referred to a specialist. The patient’s medical history revealed the absence of any systemic disorders or allergies. The dental history revealed trauma to the maxillary anterior region in childhood and subsequent gradual discolouration of the maxillary left central incisor. The patient presented with no complaints. Clinically, there were no pain or sensitivity to percussion or palpation. Tooth mobility was not increased. The tooth was restored by the referring dentist with a temporary restoration (composite resin) and showed no response to the pulp sensibility test (cold test). The radiograph and CBCT images showed that the endodontic treatment was initiated with wide access in the coronal part and confirmed root canal obliteration (Figure 16). After discussing the high probability of perforation and further tooth substance loss, an endodontic guide was designed.

##### Three-Dimensional Treatment Plan

The endodontic guide was planned as in the previous case, using the combined images of CBCT (CS 8100 3D, Carestream, 50 × 50 mm) and the intraoral scan uploaded into the software for virtual planning (Blue Sky Bio). The top of the sleeve was 20 mm from the radiographically visible part of the root canal. The guide sleeve was 6 mm long and 1 mm in diameter (Figure 17).

##### Canal Treatment

The correct fit of the guide was verified before and after the rubber dam was placed. The temporary filling was removed, and the previous access was checked with the guide, which revealed an almost correct path with no need to fill the created space. The Munce Discovery Bur #1, with a diameter of 0.8 mm (ISO head size 08), a length of 34 mm, and a speed of 10,000 rpm, was used through the guide for the first 2 mm. Due to the combined heights of the long coronal part and the guide sleeve, it was impossible to reach the demanded access with the bur. A 31-mm-long #10 K-file (VDW, Munich, Germany) was used through the guide to reach the desired length. The guide was removed every 2 mm to rinse the cavity and control the root path using the optical microscope. The canal was located at a length of 18 mm from the top of the guide sleeve—3 mm closer than the virtually planned depth. The working length was established using an electronic apex locator. A conventional root canal treatment followed (Figure 18). The time needed to localize the canal path in this case was approximately 10 min. 

### 2.2. Canal Calcification in Elderly Patients

#### 2.2.1. Case Report—Patient C: Asymptomatic Pulp Canal Obliteration—Endodontic Treatment before Surgical Procedure

A 72-year-old female patient was referred to an endodontic specialist for the treatment of the maxillary left first premolar (tooth 24) before the surgical procedure of removing a periapical lesion. The endodontic treatment had been initiated by the referring dentist, but it was impossible for the clinicians to localize the buccal root canal under the optical microscope. The medical history revealed controlled diabetes. The patient presented with no complaints. Clinically, there were no pain or sensitivity to percussion or palpation and tooth mobility was not increased. The tooth was restored by the referring dentist with a temporary restoration (composite resin) and showed no response to the pulp sensibility test (cold test). The radiograph and the CBCT (CS 8100 3D, Carestream, 50 × 50 mm) images confirmed root canal obliteration (Figure 19). After discussing a high probability of perforation and further tooth substance loss, an endodontic guide was designed.

##### Three-Dimensional Treatment Plan and RCT

The guided endodontics demanded more complex planning in this case due to artefacts caused by metal restoration in a neighbouring tooth (the second premolar, tooth 25) and the completely nonvisible root canal. According to Buchgreitz et al., when CBCT does not allow clinicians to visualize the canal, the target point in single-rooted teeth can be established through the centre of the root, as seen in the axial view [25]. Despite the fact that tooth 24 is not a single-rooted tooth, in this case, the virtual drill orientation was defined through the centre of the buccal root. A margin of 2 mm to ensure sufficient root dentine thickness was preserved, as there was no visible part of the root canal. The top of the sleeve was 15.5 mm from the bottom of the tooth chamber. The guide sleeve was 6.5 mm long and 1 mm in diameter (Figure 20). The teeth-supported guide was designed with an embossed canal marking (Figure 21).

The buccal canal was cautiously negotiated through the guide with a size 10 file (C-Pilot #10) instead of rotated burs (Figure 22). The root canal orifice was reached at a length of 17 mm from the top of the guide sleeve. The time needed to localize the canal path in this case was approximately 15 min. The file was taken to the working length and a conventional root canal treatment followed (Figure 23).

After finishing RTC, the patient was referred to a dental surgeon specialist for further treatment.

#### 2.2.2. Case Report—Patient D: Symptomatic Pulp Necrosis with Pulp Canal Obliteration

A 68-year-old female patient reported to the endodontic specialist complaining of acute pain upon percussion of her maxillary first premolar (tooth 24), experienced for several days. The patient was healthy and did not have any general or chronic diseases. The clinical examination revealed tenderness to percussion and a negative response to the pulp sensibility test (cold test). Radiographic images revealed pulp canal calcification and the absence of the canal light (Figure 24). A CBCT scan was performed (CS 8100 3D, Carestream, 50 × 50 mm) and confirmed canal calcification (Figure 25). The palatal canal was found during the first visit. The buccal canal was obliterated. Due to the possible risk of higher tooth substance loss and perforation during endodontic access, we decided to perform the endodontic treatment with guided access.

As in the previous case, the virtual drill orientation was established in the centre of the root. The top of the sleeve was 14 mm from the bottom of the tooth chamber. The guide sleeve was 6 mm long and 1 mm in diameter (Figure 26). The template with engraved canal markings was equipped with a window for assessing the correctness of the guide’s intraoral placement (Figure 27). The buccal canal was cautiously negotiated through the guide with a size 10 file (C-Pilot #10) instead of rotated burs. The root canal orifice was reached at a length of 15 mm from the top of the guide sleeve. The time needed to localize the canal path in this case was approximately 10 min. The file was taken to the working length and a conventional root canal treatment followed (Figure 28).

## 3. Discussion

Root canal treatment in teeth with calcified canals should be performed if the tooth presents radiological signs of periapical disease or if symptoms are reported by the patient (e.g., pain, tenderness to percussion, etc.)—or before prosthetic or surgical treatment. The endodontic procedure should be carefully planned and performed to avoid deviation from the original canal path, root perforation, and excessive loss of tooth structures. Severe calcification or a complex anatomy of the tooth may pose a problem or induce stress, even for a very experienced clinician. The guided endodontic technique has been reported by some authors as beneficial when localizing calcified canals [14,15,18,19,25,26]. Moreno-Rabie et al. [25] performed a systematic review of the studies on the clinical application, accuracy, and limitations of guided endodontics, which revealed that this method is highly accurate and successful, though there is a need for further research. In the case reports presented in this paper, the guided endodontic technique allowed the clinicians to perform a root canal treatment in a minimally invasive manner, protecting the tooth structure. The time needed to localize the canal path was reduced to 10 min in Patients B and D and to 15 min in Patient C (time measured from the administration of local anaesthesia to the confirmation of the path with the electronic apex locator). In Patient A, the time was longer as it was the first learning case, and the procedure took approximately 50 min. The reduced amount of time in cases B–D could have also been attributable to the previously prepared chamber access. Buchgreitz et al. reported that the previous attempt to negotiate the root canal led to the greatest number of optimal precision scores following drill path preparation [24]. Even though the time needed for planning may be perceived as a disadvantage, it eventually significantly reduces treatment time and stress. The STL file of the intraoral scan and the DICOM files of CBCT are needed to design an endodontic guide. These files must be combined in a dedicated software product, e.g., Blue Sky Plan. It is important to separate the lower and upper teeth when performing CBCT examinations in order to make DICOM and STL matching possible. Planning with the use of the multi-sectional view makes it possible to access the root canal and avoid the degradation of the incisal edge or other anatomical structures of the tooth by changing the orientation of the long axis of the virtual implant. However, it is important to bear in mind the possible inaccuracy of CBCT examinations and plan the pathway for the endodontic instruments with a margin ensuring sufficient root dentine thickness [24]. The longer the guide sleeve, the lower the risk of a lateral deviation of the endodontic access tool near the apex of the virtual implant. Resin is a material with greater flexibility than metal, so the use of metal sleeves may also increase the precision of the access preparation. Metal sleeves are recommended especially in cases with thin root canal dentine walls. Placing a metal sleeve and using an adapter bur may protect the resin guide. During the treatment, multiple teeth must be isolated at once with a rubber dam to achieve guide stability. Due to the short, thick crowns, placing the guide with the rubber dam was much more difficult in Patient A than it was in Patient B, and the guide had to be shortened to achieve better stability. The described technique was limited to straight parts of root canals. Difficulties with using a guide technique may occur in patients with restricted mouth opening or in posterior teeth [2]. Exclusion criteria may also include metal restorations in the adjacent teeth, as this leads to CBCT artifacts in the area of interest [24]. The main disadvantage of guided endodontics may be the additional cost of procedures and materials such as CBCT, 3D printers, and resin. The guided technique can also be used when removing fiberglass posts [25,27,28] or during the treatment of teeth with developmental anomalies, e.g., dens invaginatus [29,30] or dentin dysplasia [31]. In the four presented cases, all calcified root canals were properly treated using endodontic guides. The CBCT examinations were performed according to the guidelines of the European Society of Endodontology [32]. A summary of the procedure is shown in Table 1.

## 4. Conclusions

Guided endodontics represents an advanced therapy that enables clinicians to perform endodontic treatment of teeth with PCO in a more predictable and precise manner and allows clinicians to avoid over-preparation and iatrogenic damage. The time needed to locate canals can be reduced. Further research may extend the number of indications for the use of guided techniques and improve digital dentistry.

## Figures and Tables

**Figure 1 ijerph-19-09958-f001:**
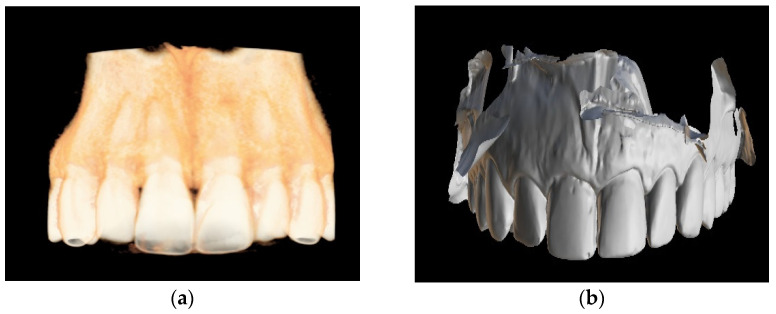
(**a**) Cone-beam computed tomography (CBCT) image presenting patient’s maxilla; (**b**) patient’s intraoral scan of upper arch.

**Figure 2 ijerph-19-09958-f002:**
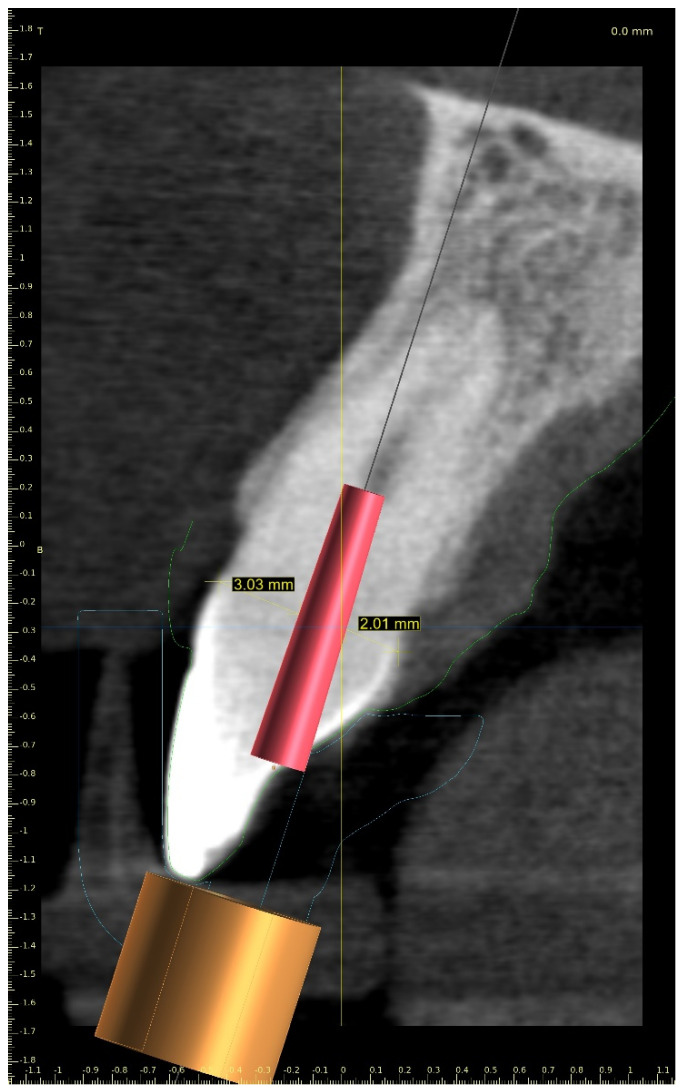
CBCT image showing visible part of root canal. Custom virtual implant was placed according to access path preserving the minimum thickness of the dentine.

**Figure 3 ijerph-19-09958-f003:**
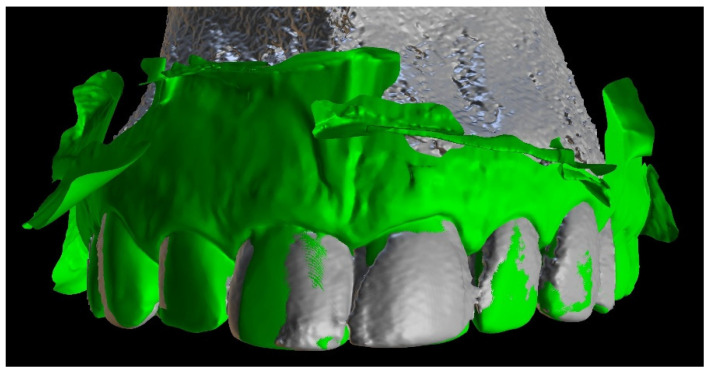
Image presenting aligned files of patient’s CBCT data and intraoral scan.

**Figure 4 ijerph-19-09958-f004:**
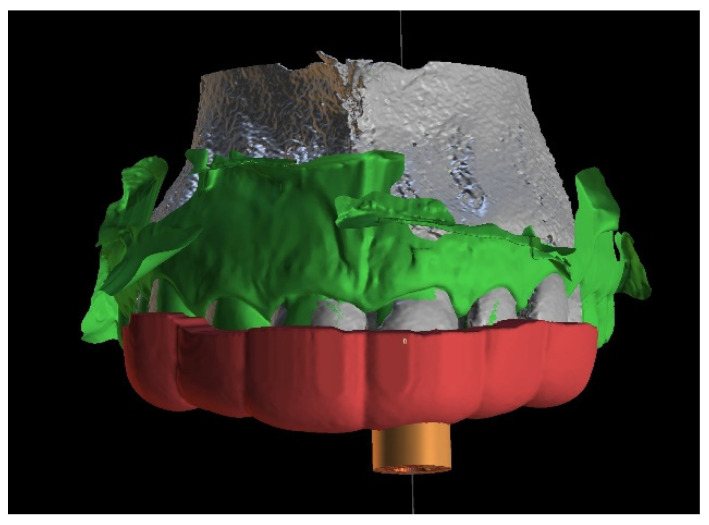
Figure showing extent of the guide.

**Figure 5 ijerph-19-09958-f005:**
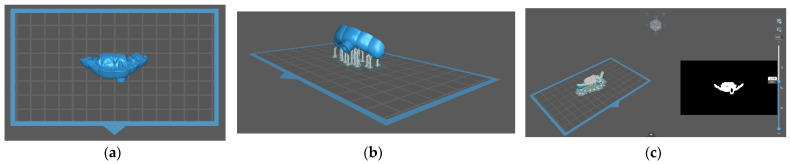
(**a**) STL file of the guide imported to the slicer software (Chitubox); (**b**) image after adding connectors for printing; (**c**) software showing object sliced into 50 µm layers. Each layer can be displayed and checked.

**Figure 6 ijerph-19-09958-f006:**
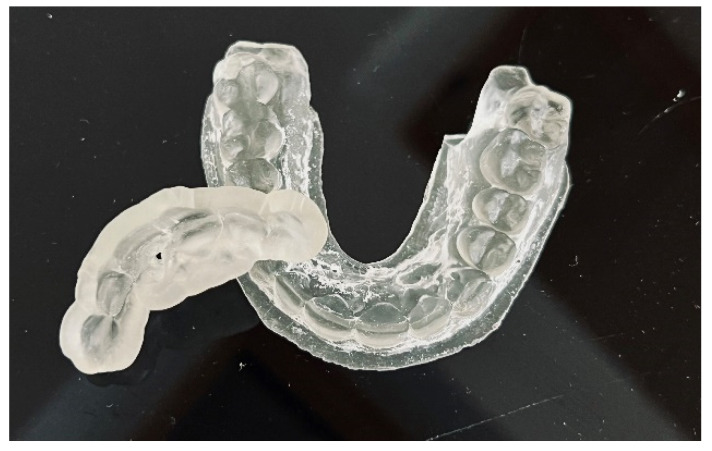
Endodontic guide and patient’s upper arch 3D-printed model.

**Figure 7 ijerph-19-09958-f007:**
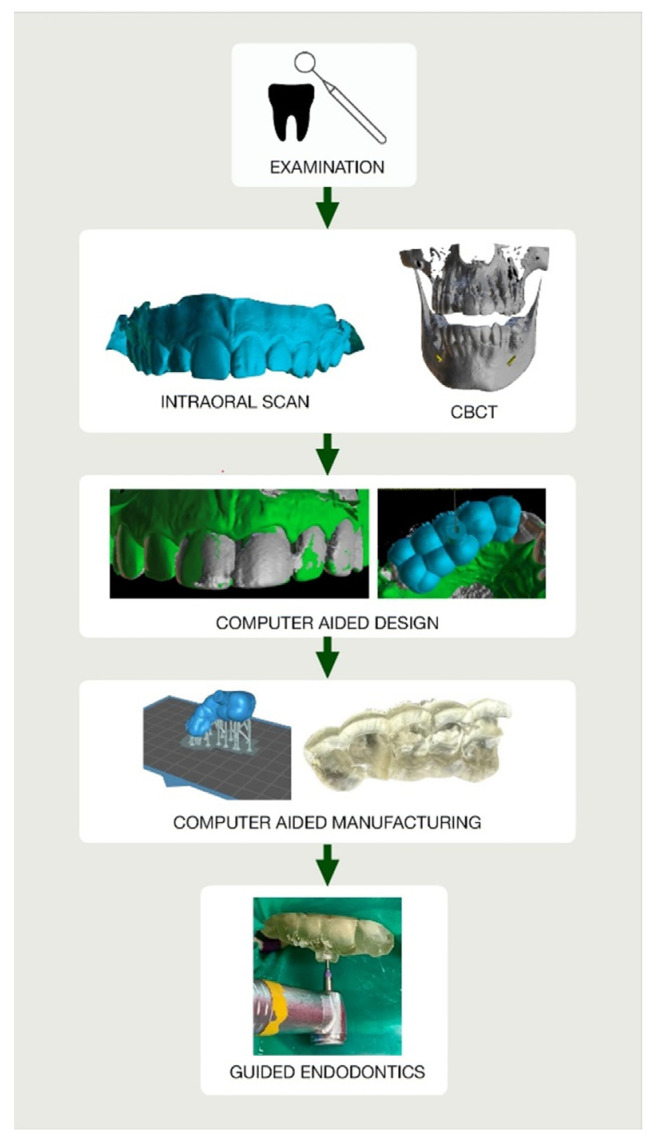
Flowchart illustrating the entire digital workflow’s steps to deliver endodontic guide.

**Figure 8 ijerph-19-09958-f008:**
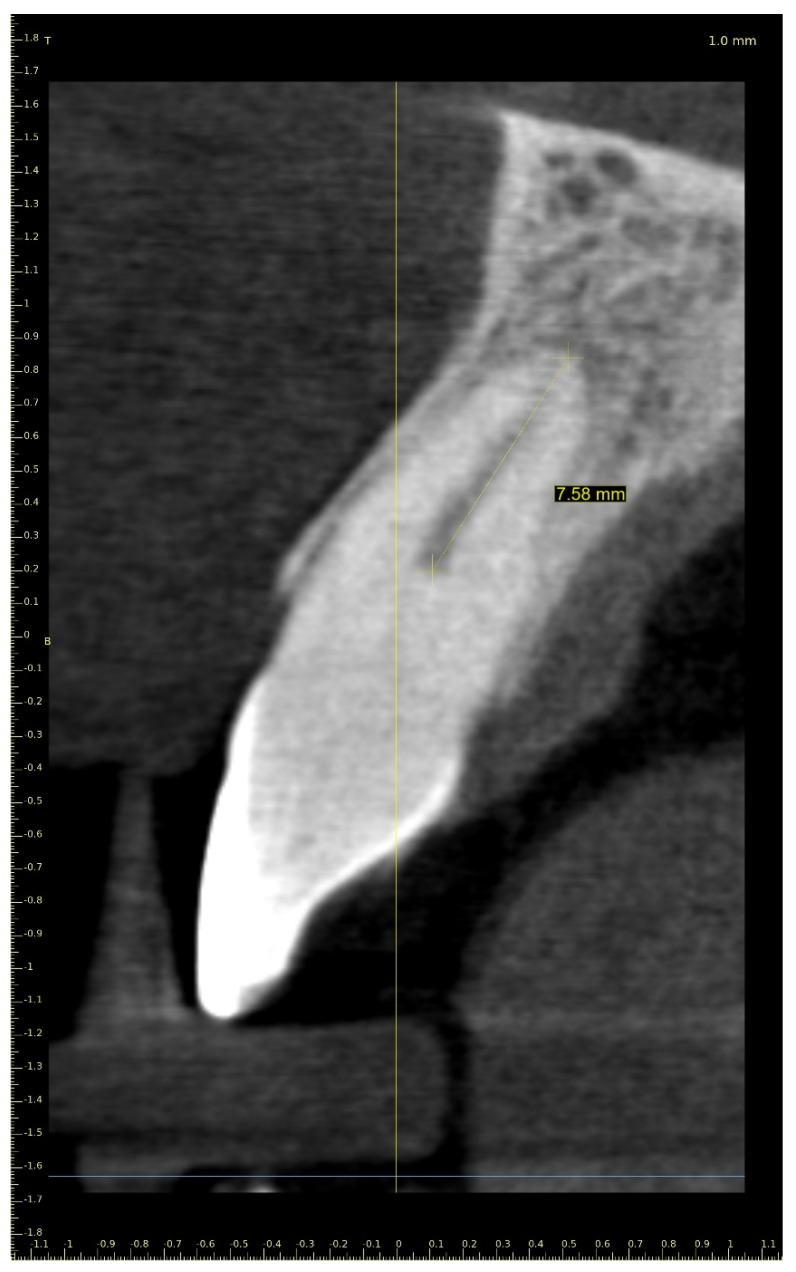
CBCT sagittal view image revealed a 7.58-mm-long visible part of the root canal on tooth 21 (FDI).

**Figure 9 ijerph-19-09958-f009:**
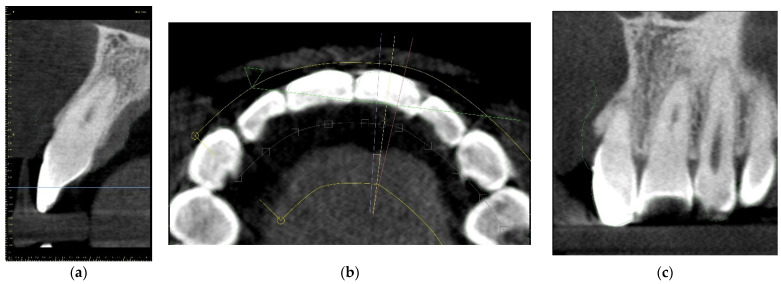
Figure showing CBCT data and intraoral scan (green line) alignment from different views: (**a**) sagittal; (**b**) axial; (**c**) coronal.

**Figure 10 ijerph-19-09958-f010:**
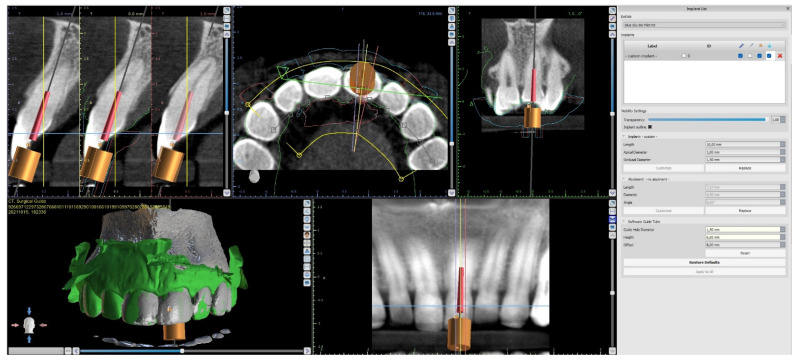
Figure presenting settings of custom implant in Blue Sky Plan software. Implant length, guide tube height, and offset must be equal to access tool’s working length.

**Figure 11 ijerph-19-09958-f011:**
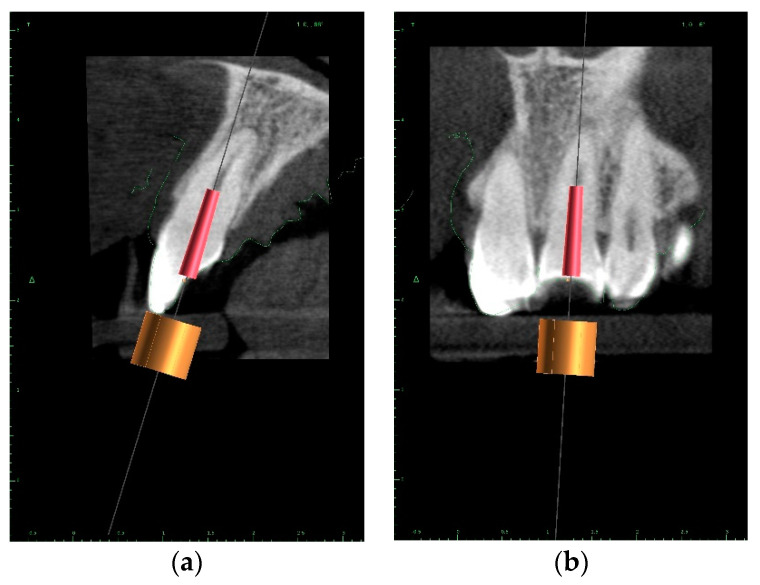
CBCT image presenting a scheduled virtual implant: endodontic access path in (**a**) sagittal view and (**b**) coronal view. The volume of the preserved dentin around the access path is shown.

**Figure 12 ijerph-19-09958-f012:**
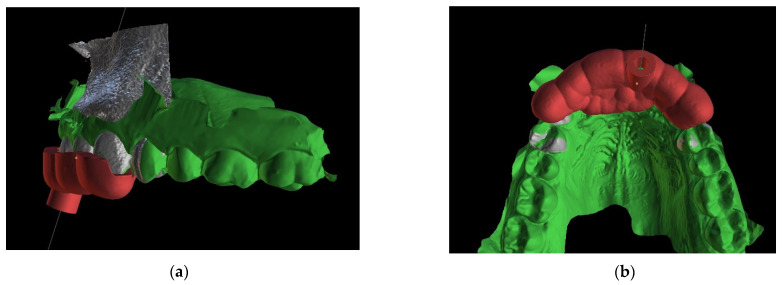
The range of the guide is drawn on the combined CBCT and STL image. This figure shows the extent of the template (**a**) labially and (**b**) palatally.

**Figure 13 ijerph-19-09958-f013:**
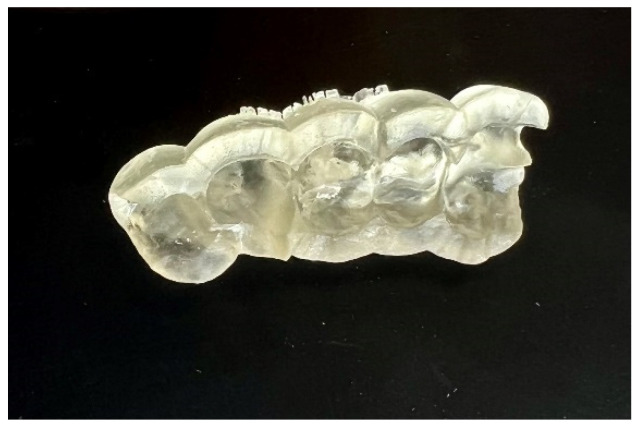
Three-dimensional printed endodontic guide.

**Figure 14 ijerph-19-09958-f014:**
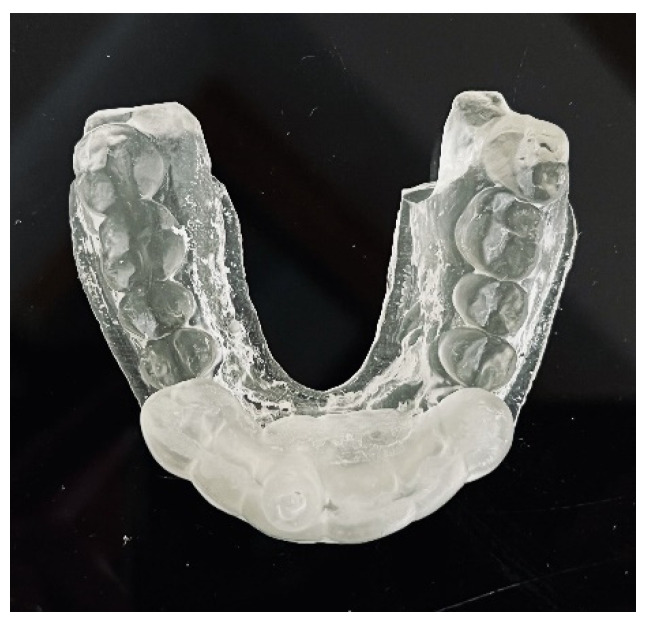
The three-dimensional printed guide and the patient’s model enabled us to check the accuracy of the guide and to practise proper drill insertion.

**Figure 15 ijerph-19-09958-f015:**
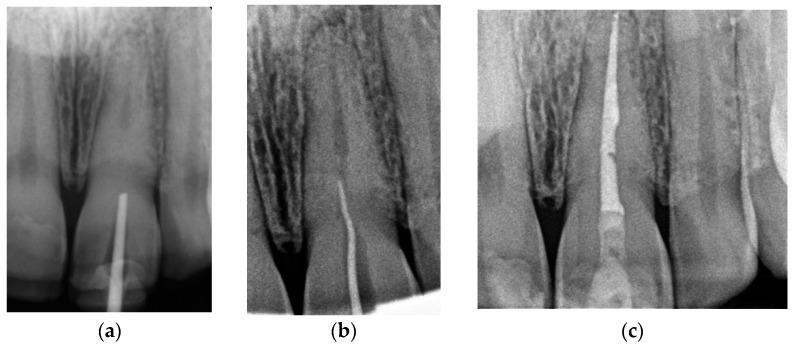
Radiographic examination was used to confirm (**a**) correct path of access pathway during procedure and (**b**) correct root canal access; (**c**) post-obturation radiograph.

**Figure 16 ijerph-19-09958-f016:**
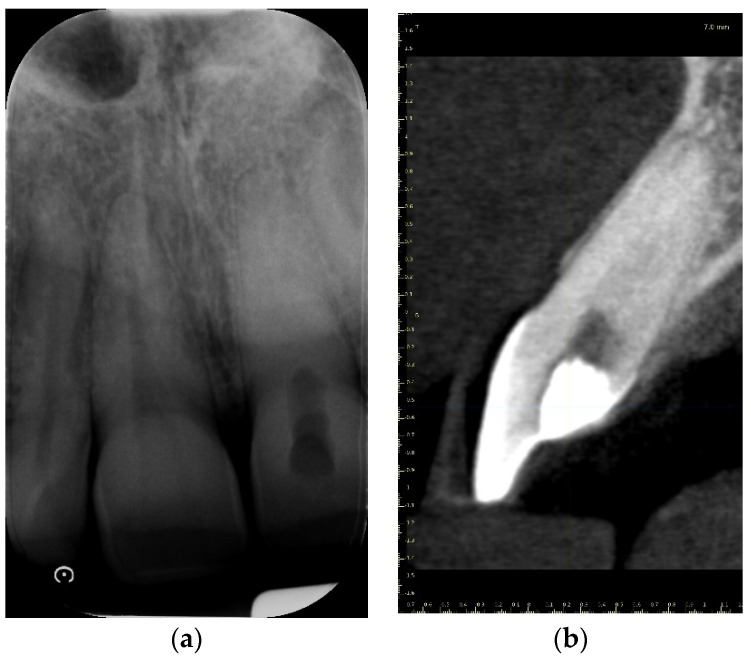
The failed wide access in the coronal part and root canal obliteration on (**a**) X-ray and (**b**) CBCT images.

**Figure 17 ijerph-19-09958-f017:**
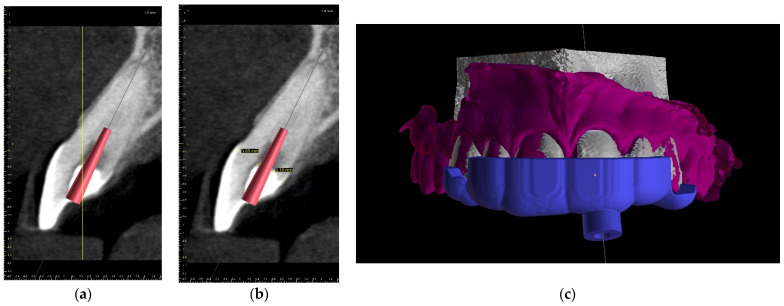
CBCT image showing (**a**) scheduled virtual implant position; (**b**) preserved tooth’s hard tissues in pericervical area around access pathway; (**c**) the extent of the endodontic guide that allowed us to place the rubber dam.

**Figure 18 ijerph-19-09958-f018:**
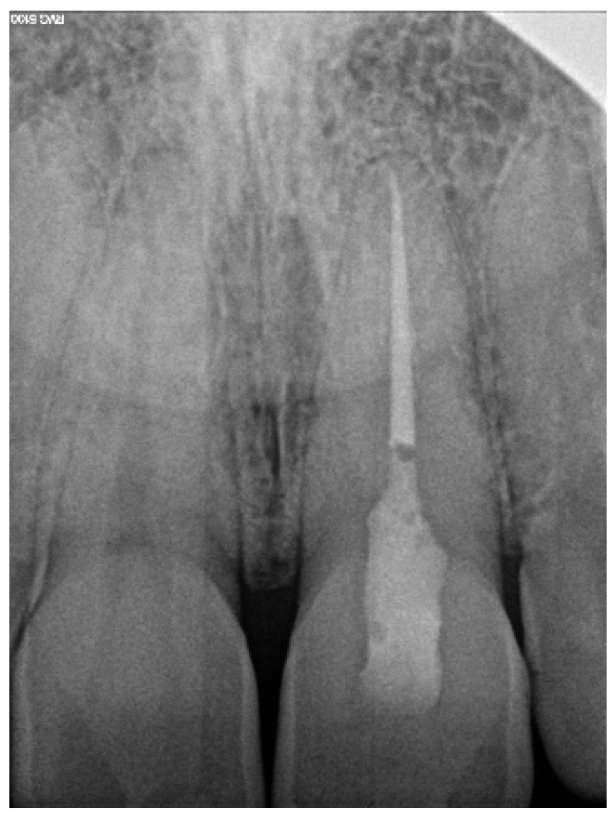
X-ray image showing the correctness of the performed root canal treatment.

**Figure 19 ijerph-19-09958-f019:**
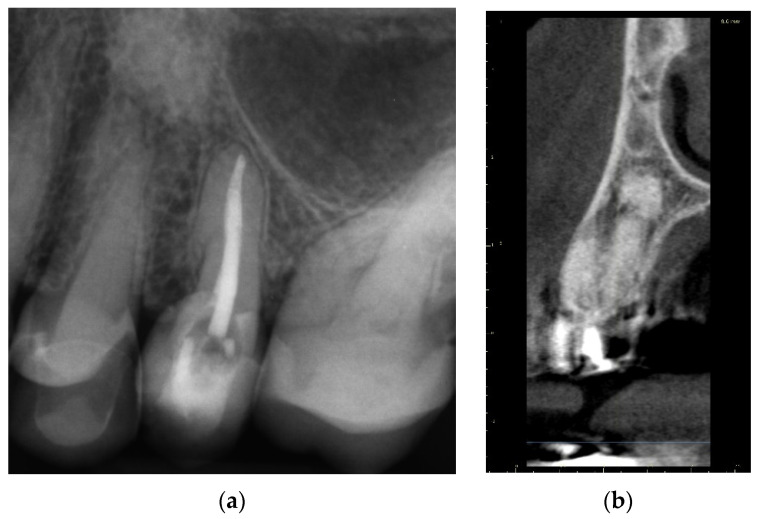
Nonvisible root canal system on (**a**) X-ray and (**b**) CBCT images (tooth 24).

**Figure 20 ijerph-19-09958-f020:**
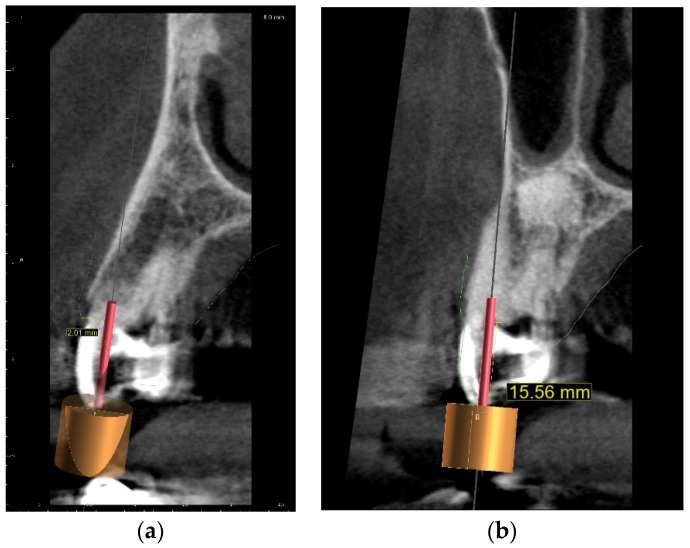
CBCT image presenting (**a**) virtual implant positioned through the centre of the buccal root, preserving a safe amount of the tooth’s hard tissues; (**b**) guide tube designed to reach the bottom of the chamber after 15.5 mm. In the lateral dentition area, short guide tubes may ensure more space for the drill.

**Figure 21 ijerph-19-09958-f021:**
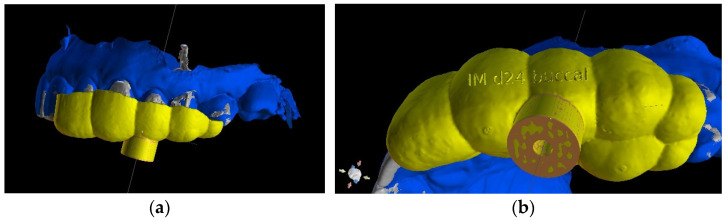
Three-dimensional image of (**a**) endodontic guide range and (**b**) embossed canal marking on the labial template’s surface.

**Figure 22 ijerph-19-09958-f022:**
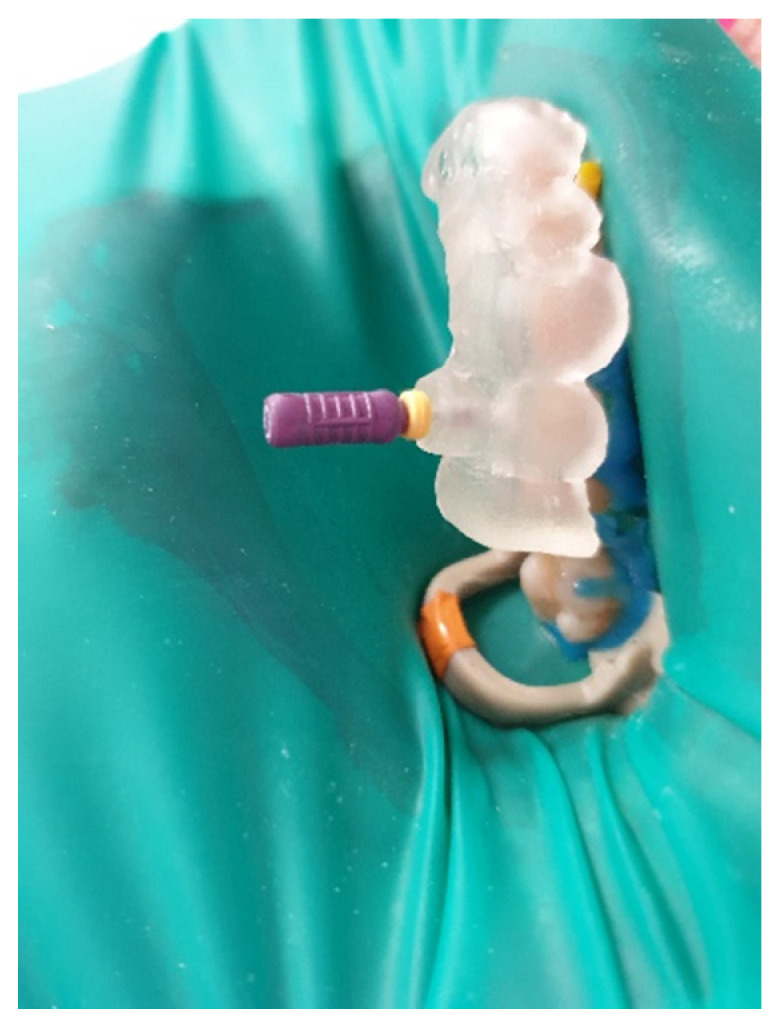
Image presenting endodontic guide placed on patient’s teeth. Root canal was negotiated with a C-Pilot file.

**Figure 23 ijerph-19-09958-f023:**
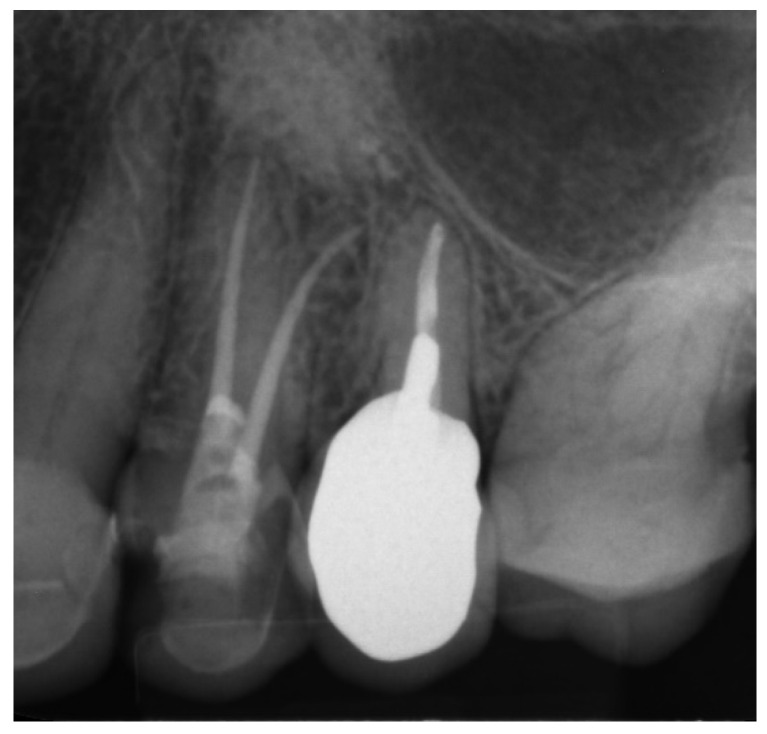
X-ray image showing the correctness of the performed root canal treatment.

**Figure 24 ijerph-19-09958-f024:**
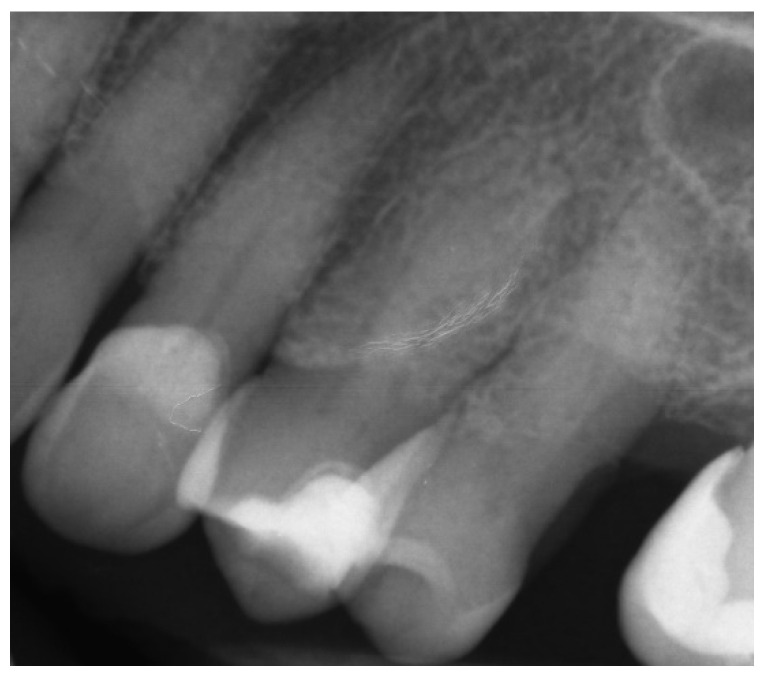
X-ray image showing the obliteration of the root canal.

**Figure 25 ijerph-19-09958-f025:**
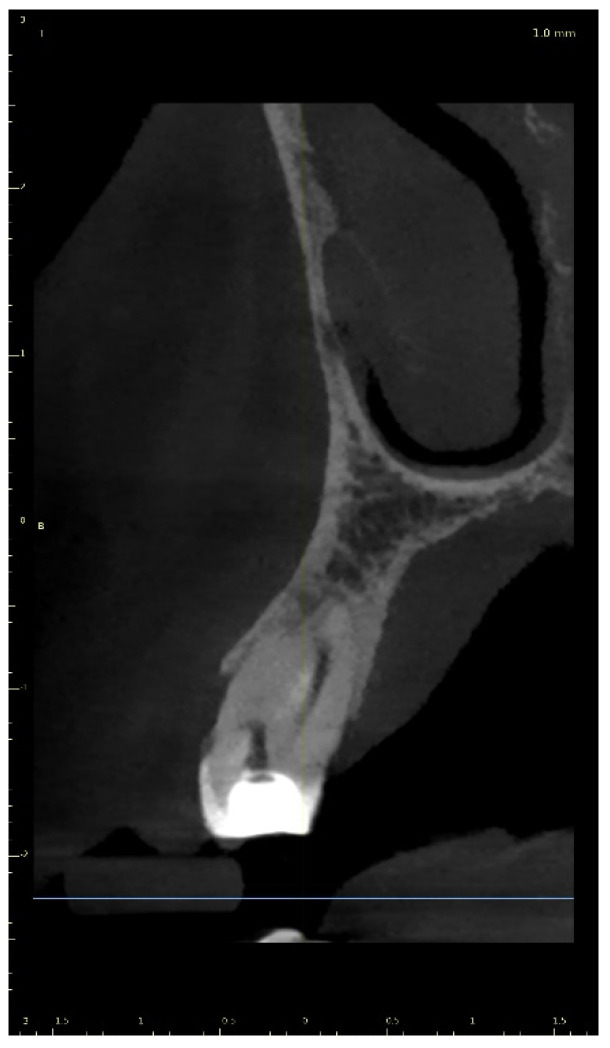
The CBCT image confirmed the calcification of the buccal canal.

**Figure 26 ijerph-19-09958-f026:**
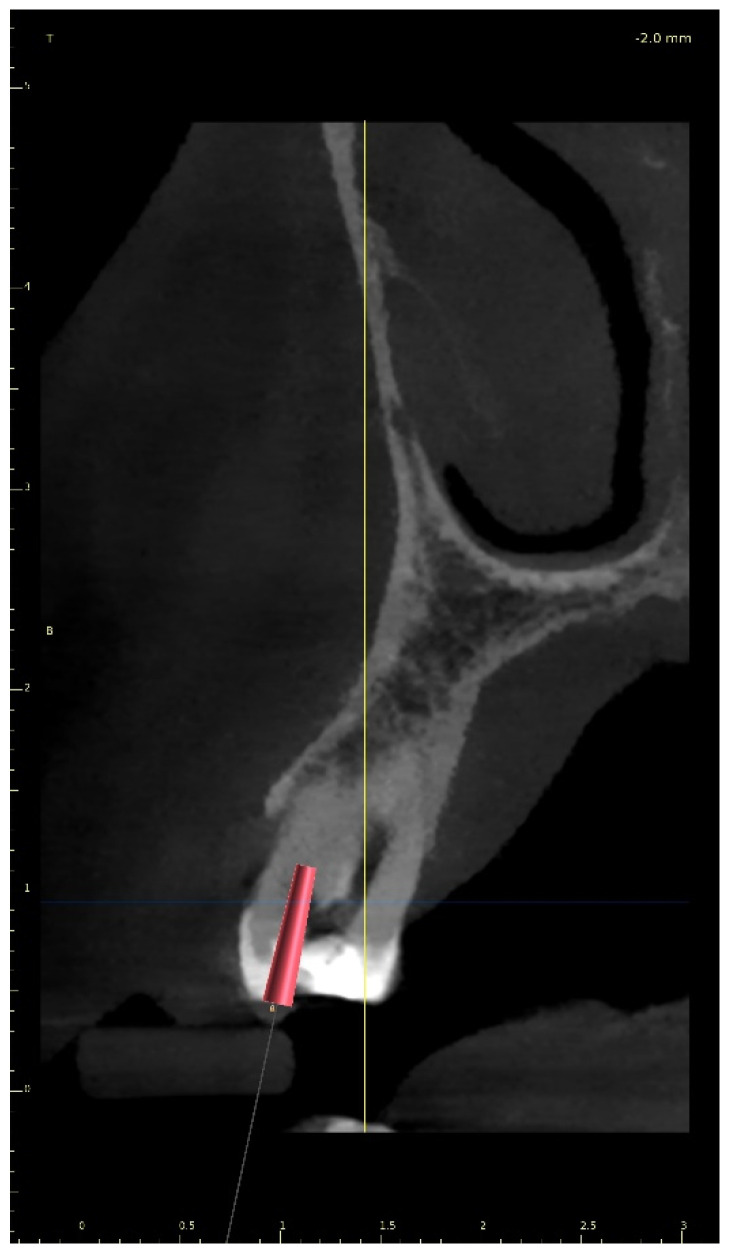
CBCT image presenting virtual implant positioned through the centre of the buccal root, as canal was nonvisible.

**Figure 27 ijerph-19-09958-f027:**
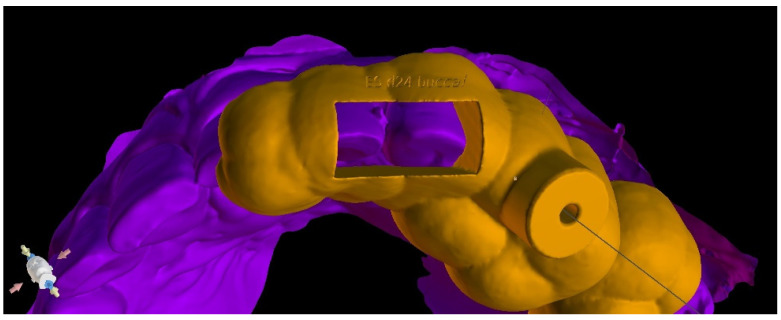
Image presenting guide with window on the occlusal site that allowed us to check the correctness of the endodontic guide placement.

**Figure 28 ijerph-19-09958-f028:**
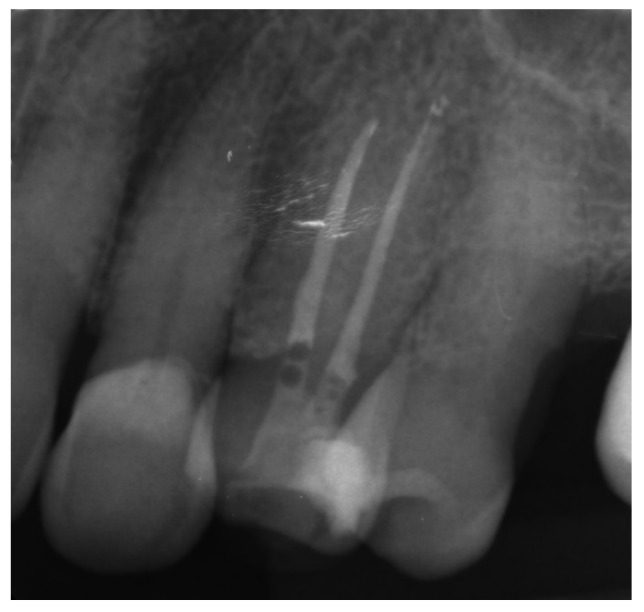
X-ray image showing the correctness of the performed root canal treatment.

**Table 1 ijerph-19-09958-t001:** Digital and clinical workflow of guided endodontics.

Stage
Examination
Cone-beam computed tomographic scan
Digital intraoral impression: Directly—intraoral scan Indirectly—scanning impression or plaster model
Import DICOM and STL files into digital planning software
Design the virtual drill path and the endodontic guide
Three-dimensional printing
Control the fit of the guide before and after placing rubber dam
Make a sign through the guide to indicate the access point in non-treated teeth
Remove the enamel until dentine is exposed
Place the guide on the teeth
Work through the guide: Use rotate burs in dentine, scout the canal through the guide
Remove the guide to rinse the cavity and clean the burs, control endodontic access using an optical microscope
Perform a radiographic examination to confirm correct canal access
Complete the root canal treatment

## Data Availability

Not applicable.

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
