# Peer review of "Guided Endodontics as a Personalized Tool for Complicated Clinical Cases"

_ijerph, 2022, doi:10.3390/ijerph19169958_

Round 1

Reviewer 1 Report

The study is well conducted. Table 1 is probably not necessary because all the steps are well described in the test.

Perhaps the only downside to this method is the high cost of everything you need.

I have some doubts about the usefulness of reference n. 13

Author Response

Dear Reviewer,

We are very grateful for your insightful comments on our paper and appreciate the time and effort that you have dedicated to provide your valuable feedback on our manuscript.

Here is a point-by-point response to your comments.

Table 1 is probably not necessary because all the steps are well described in the test.

>> We agree with the reviewer that all steps and general workflow is described in the text. However, we believe that it is useful for general clinician to sum up step-by-step procedure. For this reason, we chose not to remove it, however we have modified it to shorter version – Page 17, line 433.  –

Perhaps the only downside to this method is the high cost of everything you need.

>>  Agree. We have done comment to emphasize this point – Page 17, line 426

I have some doubts about the usefulness of reference n. 13

>>  Thank you for pointing this out. We apologize for our error. We’ve changed these reference n.13  to:

Fernandes T. O.; Abreu M. G. L.; Antunes L. S.; Antunes L. A. A. Factors associated with pulp canal obliteration due to traumatic injuries in deciduous teeth: a retrospective study. Int J Burns Trauma 2021, 11(4), 304-311. PMID: 34557333; PMCID: PMC8449148.

Revised manuscript is attached.

We would like to thank again for taking the time to review our manuscript.

Reviewer 2 Report

Dear Authors,

Thank you for the opportunity to review your work.  It is very interesting.  However, there is a need for extensive revision for grammar and English.  I began to make suggestions, but there are too many corrections that were needed to do so.  

Suggested changes ABSTRACT

 The aim of the study is to present a technique to individualize root canal localization in teeth with calcified root canals using digitally planned and 3D-printed endodontic guide.

Merging CBCT images and an intraoral scan allows a clinician to prepare an endodontic guide.

 The template was printed on a 3D printer using transparent 20 resin.

 Navigated endodontics enables clinicians to perform

Suggested changes INTRODUCTION

 For the last three decades, we have seen many technological advancements.  Among the new advancements used in clinical dentistry are 3D printers and 30 cone – beam computed tomography (CBCT) technology.

Among the treatment objectives of root canal treatment (RCT) is the need to preserve normal periradicular tissues.

 After evaluating the medical and dental histories, determining the chief complaint, and diagnosing a need for RCT, the clinician will need to properly access the tooth.

Preserving the pericervical dentin functions as a stress distributor.  The pericervicl dentin may improve the resistance to the fracture and increase the probability of success with a future prosthetic restoration.

Use PCO in line 53 without the definition, as the definition was already presented in line 49.

Author Response

Dear Reviewer,

We are very grateful for your insightful comments on our paper and appreciate the time and effort that you have dedicated to provide your valuable feedback on our manuscript.

We agree with your suggestions about English grammar and style. We have revised the text and  incorporated your suggestions throughout the manuscript. We hope that it is now clearer. Changes can be seen in “Track Changes” mode. Revised manuscript is attached.

We would like to thank again for taking the time to review our manuscript.

We will be grateful for further suggestions.

Reviewer 3 Report

This manuscript is an introductory study, it has to be at the clinical trial level without a control group.

 The resolution of CBCT is also a problem, and the clearer the CT image, the more accurate the guide stent will be.

 More specific research is considered to take more time, and I think that this article is maintained at an appropriate level in its current state. 

Overall, it's well written and I think it might be helpful to clinicians.

Author Response

Dear Reviewer,

We are very grateful for your insightful comments on our paper and appreciate the time and effort that you have dedicated to provide your valuable feedback on our manuscript.

We totally agree that in future cases the CBCT should be performed in clearer way to improve guide’s accuracy and thank you for pointing this out. As well as further research should be provided at the clinical trial level.

Revised manuscript is attached.

We would like to thank again for taking the time to review our manuscript.

Round 2

Reviewer 2 Report

Dear authors,

The beginning of the manuscript has been greatly improved.  There is still a need for moderate improvement in the editing of the middle and end of the manuscript.  

I believe the manuscript warrants help by the editor for grammar.  I am accepting with minor revisions to grammar/text editing.

Author Response

Dear Reviewer,   We greatly appreciate the thorough comments provided on our manuscript.

We agree with your suggestions about English grammar and style. Manuscript has undergone English language editing by MDPI. The text has been checked for correct use of grammar and common technical terms and edited to a level suitable for reporting research in a scholarly journal by English editor - Gareth Thomas. Changes can be seen in “Track Changes” mode. Revised manuscript is attached.

We would like to thank again for taking the time to review our manuscript.

We will be grateful for further suggestions.
